# GC Insights: Breaking the silos – leveraging NLP to encourage interdisciplinary interaction at the EGU

Jan Sodoge[1], Taís Maria Nunes Carvalho[1,2], and Mariana Madruga de Brito[1]

[1]Department of Urban and Environmental Sociology, UFZ-Helmholtz Centre for Environmental Research, 04318, Leipzig, Germany
[2]Center for Scalable Data Analytics and Artificial Intelligence (ScaDS.AI) Dresden/Leipzig, Universität Leipzig, Leipzig, Germany

**Correspondence:** Jan Sodoge (jan.sodoge@ufz.de)

**Abstract.** Thousands of abstracts from various geoscience sub-fields are presented annually at the EGU General Assembly (GA), offering a rich resource for tracking scientific progress. However, rigid session groupings can limit cross-disciplinary exploration. Here, we show that participants focusing only on their broad disciplinary divisions miss an average of 44 % of the 10 most relevant contributions. To break this compartmentalization, we propose using natural language processing (NLP), enabling the geoscience community to explore the full breadth of knowledge beyond traditional disciplinary boundaries.

## 1 Introduction

Each year, the European Geosciences Union (EGU) General Assembly (GA) gathers over 20,000 geoscientists worldwide, with participation steadily increasing since its inception in 2004 (EGU, 2024). To organize this vast array of research, the abstracts are organized into 22 disciplinary programme groups (PGs), alongside Inter- and Transdisciplinary Sessions (ITS), Education and Outreach Sessions (EOS), and other Union-wide Sessions. We hypothesize that this structure may inadvertently create knowledge silos, as EGU GA attendees might tend to focus on their own scientific programme groups —also referred to as divisions[1]-potentially missing relevant developments from other disciplines. As a result, they may only be exposed to ideas within their peer group, thereby reinforcing existing perspectives. This phenomenon mirrors the well-documented effects of filter bubbles and selective exposure to information observed on social media platforms (Robertson et al., 2023; Spohr, 2017). In scientific settings, such bubbles can hinder interaction between fields that could catalyze creativity and innovation (Kittur et al., 2019; Burt, 2004). While the idea of bursting such potential bubbles is thought to foster research progress (Portenoy et al., 2022), this phenomenon remains underexplored in the context of large scientific conferences, such as the EGU GA.

Recent advancements in natural language processing (NLP) offer a promising solution to address this compartmentalization. Language models can extract structured insights from vast amounts of scientific documents, thereby revealing knowledge that would otherwise remain hidden within the sheer volume of scientific output (Yau et al., 2014; De Battisti et al., 2015; Chen et al., 2019; Sodoge and de Brito, 2024). For example, language models have been successfully used to map climate-change-related publications (Callaghan et al., 2020), moving towards understanding the meaning and context of language.

---

[1]We use the terms "divisions" throughout the text for clarity

In this work, we investigate how the compartmentalization at the EGU GA programme may generate filter bubbles. We then demonstrate how NLP can be used to organize knowledge from EGU GA abstracts beyond traditional disciplines, benefiting conference participants, organizers, and the broader geoscience community. Using the abstracts from the EGU GAs (2020-2024), we create a *textual cartography* of the geoscience landscape, aiming to: (1) provide an overview of geoscience research presented at the EGU GAs, (2) guide participants to relevant research across disciplinary boundaries, and (3) assist in organizing conference sessions.

## 2   Methods and Data

Our approach consists of 5 steps, including: (1) collecting abstracts from EGU, (2) computing similarities between abstracts, (3) visualizing the abstracts in a 2-dimensional space, (4) running a simulation to estimate the hypothesized filter effect, and (5) comparing the coherence of divisions created through clustering to the existing division structures.

To demonstrate our approach, we collected abstracts from the past five EGU GA. For each GA, we scraped the abstracts and corresponding division. After removing withdrawn abstracts, the dataset comprised 77,911 contributions across 22 disciplinary divisions (2020: 17728; 2021: 13368; 2022: 12129; 2023: 15772; 2024: 18914) and 3,663 abstracts from the Inter- and Transdisciplinary Sessions (ITS). We have not considered the ITS abstracts as a single division, as (1) their content and number of sessions may vary significantly across years, making it difficult to analyze trends consistently; (2) they represent a small fraction of total submissions (around 0.5 % of all abstracts). Instead, we reassigned the ITS abstracts to their co-organizing divisions (e.g., ITS and NH) to account for their disciplinary interlinkages.

We then used a pre-trained language model to generate a text embedding for each abstract. An embedding is a high-dimensional vector of numerical values representing the text's semantic meaning. These reveal relationships between abstracts as texts with similar content have similar embeddings. The pre-trained model requires no additional preprocessing (e.g. stop-word removal), as it is designed to ignore irrelevant parts and focus on meaningful patterns in the text. Specifically, we employed the *distilroberta-base* language model (Liu, 2019), which was trained on large text corpora, enabling it to capture text structure, meanings, and context. This model was selected for its ability to capture the entire abstract and robust performance across diverse language tasks (Naseer et al., 2021; Briskilal and Subalalitha, 2022).

In order to visualize the text embeddings and create a textual cartography of geoscience research, we applied the U-MAP (Uniform Manifold Approximation and Projection) dimensionality reduction method (McInnes et al., 2018). This technique projects the high-dimensional text embeddings onto a 2-dimensional space while preserving the local and global structure of the data. The result is a map where similar abstracts are positioned closer and distinct content farther apart.

To investigate the hypothesized filter bubble effect, we conducted a simulation by randomly selecting 5,000 abstracts from each EGU GA. For each abstract, we identified the 10 most similar abstracts presented in the same year using cosine similarity (Kenter and De Rijke, 2015). We then analyzed the disciplinary divisions associated with these similar abstracts to calculate the proportion that belonged in the same versus different divisions. A higher proportion of similar abstracts from different

divisions suggests that EGU GA participants may miss out on potentially relevant contributions by focusing exclusively on their own division.

Finally, to evaluate how well the EGU GA abstracts were grouped, we compared the clustering quality when using the disciplinary divisions against using the k-means clustering algorithm (Rodriguez et al., 2019), considering the same number of clusters as the number of disciplinary sessions (n=22). The average Silhouette coefficient (Dinh et al., 2019) was used to assess cluster coherence, where values close to 1 indicate well-separated clusters and negative values indicate no coherent cluster structures. This comparison allowed us to assess which method clusters abstracts more effectively that address similar topics.

## 3 Results

Figure 1a shows the research landscape of the EGU GA from 2020-2024, featuring a total of 77,911 abstracts. Each abstract is colored by their respective disciplinary division, highlighting the relationships among them. For example, abstracts from the natural hazard (NH) and climate (CL) divisions are spread across different clusters. A qualitative inspection of this map reveals not only such macro-level structures but also the similarities and differences on a more fine-grained level. For instance, the contributions most similar to a research abstract on modelling farmers' irrigation preferences (Heilemann et al., 2024) include an abstract evaluating irrigation demand in the same case-study area (Fallah-Mehdipour and Dietrich, 2024) and an abstract on the modelling of global irrigation water demand (Beier et al., 2024). These were presented in different divisions and thus would potentially be missed if an attendee only consulted one division and the sessions are not co-organized.

Our findings confirmed the presence of a filter bubble effect, with varying degrees across the EGU GA divisions. On average, participants who restrict their attention to abstracts within their own division potentially miss 44% of the 10 abstracts most similar to their own abstract. This percentage varies depending on the division considered. Divisions that are rather separated, such as Atmospheric Sciences (AS) and Solar-Terrestrial Sciences (ST), have a lower likelihood for attendees to miss relevant contributions (Fig. 1b). Conversely, divisions with contributions widespread across different topics, such as Geosciences Instrumentation & Data Systems (GI) and Nonlinear Processes in Geosciences (NP), reduce the share of relevant contributions covered by the own division. We found only a weak correlation (Spearman's Rho = 0.38) between the number of abstracts of each division and the previously discussed share of relevant presentations. This suggests the impact of the size of a division on a potential filter bubble effect is limited and unlikely to strongly influence our findings.

Adding the abstracts from the ITS sessions to their co-organizing divisions significantly increased the number of missed relevant contributions in the divisions of: AS, CR, BG, HS, and SSS. This may be due to the lower thematic coherence of the ITS abstracts, which makes their text less likely to align with that of the disciplinary divisions that co-organize the ITS. Yet, these findings underscore the value of the ITS sessions themselves: the described lower thematic coherence is important for bringing together contributions from diverse fields, facilitating cross-disciplinary exchange.

When comparing different ways of organizing the abstracts, we found that grouping them by disciplinary divisions yields an average silhouette coefficient of -0.17. In contrast, applying the k-means clustering algorithm results in an average silhouette

coefficient of 0.39. This suggests that using a statistical clustering approach to group abstracts produces more coherent clusters than the divisions.

## 4  Discussion

Abstracts presented at EGU GA every year provide a snapshot of current geoscientific research. Yet, the increasing number of conference abstracts, over 18,000 in 2024, makes it challenging for participants and organizers to keep track of relevant contributions. By creating a textual cartography of more than 77,000 abstracts presented in the last five GAs, we show here how NLP can help capture the broad range of subjects discussed at EGU GA, as well as identify relevant contributions outside of the field or discipline the participant identifies with.

Our results indicate the presence of a bubble effect, where EGU GA attendees may miss contributions addressing similar problems from a different perspective presented outside their own division. This effect is evident, particularly for abstracts submitted to Nonlinear Processes in Geosciences (NP) and Geosciences Instrumentation & Data Systems sessions (GI). As a result, participants who restrict their participation to their own divisions may be exposed only to content they are familiar with, due to the restrictive nature of session compartmentalization.

To tackle this, we propose an online application designed to recommend EGU GA participants' other contributions, independent of the divisions to which they were submitted. Using interactive visualizations such as the map in Fig 1a, EGU GA participants can identify relevant research contributions beyond their immediate field (e.g. addressing the same problem but using different methods, or using the same method for different problems). This tool aims to facilitate interdisciplinary exploration by surfacing relevant contributions that attendees might not discover if they restrict their participation to a particular

session or division. This can foster interdisciplinary connections leading to innovative work and higher productivity (Portenoy et al., 2022; Specht and Crowston, 2022). During EGU GA 2024, we tested a prototype of such an application (Sup. Fig. 1) and received positive feedback on the relevance of the suggested talks.

## 5  Limitations and future research

In our analysis of filter bubbles, we recognize that the assumption that GA attendees attend only sessions on their own division

is an oversimplification. In reality, participants engage with ITS and neighboring divisions. However, due to the large volume of abstracts and sessions per division (for example, in 2024, the median number of abstracts per disciplinary division was 638, with a maximum of 2,681), manually identifying all relevant contributions across divisions is infeasible. Another key limitation is that our scraped dataset includes only co-organizing groups, not the primary organizers, which may affect the results, particularly for ITS sessions, where the roles of each division vary.

Looking ahead, we plan to conduct a more in-depth comparison of models and consider fine-tuning existing approaches to improve accuracy. Also, to address these limitations, we are currently conducting a survey to more explicitly measure attendee

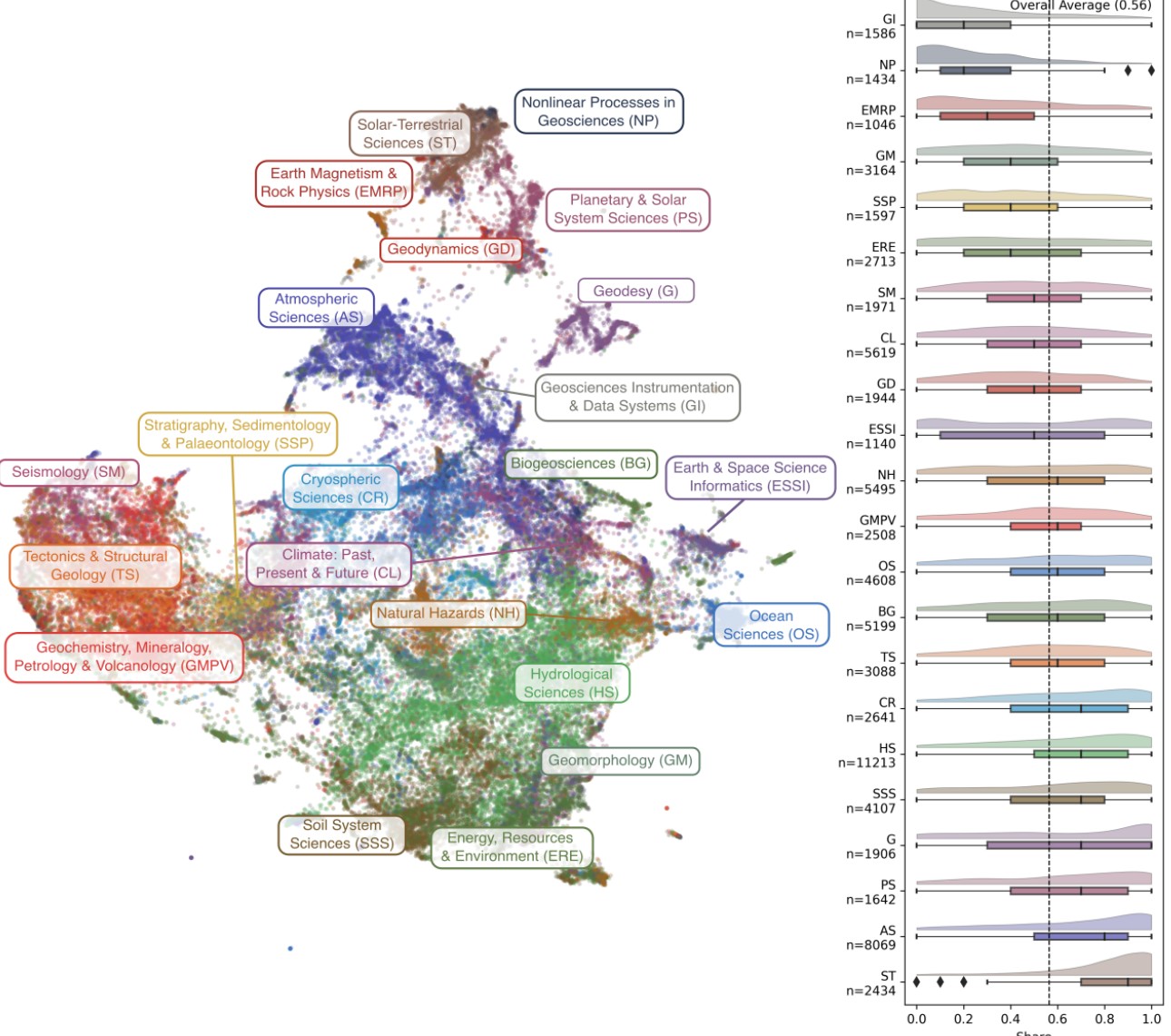

**A** Textual cartography of the geoscience landscape in the 2020-2024 EGU GAs

**B** Share of relevant contributions in the same disciplinary session

**Figure 1.** A: Textual cartography of 77,911 abstracts from the EGU GAs 2020-2024. Each dot represents an abstract, where a shorter distance between dots indicates a greater similarity in content. B: Assessment of the potential filter bubble effect due to the divisions' compartmentalization. Each observation represents a randomly selected abstract. Its value indicates the proportion of the 10 most similar abstracts in the same division as the selected abstract. Higher values suggest a stronger tendency for relevant abstracts to be concentrated within the same division. *n* denotes the total number of abstracts in each session between 2020-2024.

behavior and evaluate the assumptions underlying our analysis by incorporating user feedback. This will allow us to assess whether the suggested contributions are perceived as meaningful by EGU GA attendees.

## 6 Conclusion

We showed that clustering the abstracts from the EGU GA using k-means, compared to relying on disciplinary divisions, achieved a better clustering, i.e., more semantically coherent sessions. While these results are provocative, our intent is not to suggest replacing divisions or other formal groupings. Large conferences, such as EGU GA, require these organizational structures to coordinate and design a coherent program. We also recognize the role of authors in selecting the most appropriate session, often influenced by community ties. Hence, the computational approach suggested here is seen more as a tool in
a co-design process, helping organizers and participants make more informed decisions when faced with a vast amount of information. In fact, the authors of this study are collaborating with the EGU Programme Committee and the Copernicus Conference Manager to experiment with additional tools that can be integrated into the conference system and enhance the attendee experience.

Beyond the application provided here, the developed textual cartography opens multiple new research avenues for studying
geoscientific research. Our results suggest new possibilities for assessing trends across different geoscience subfields over time, identifying emerging research areas, and tracking shifts in scientific focus that might otherwise go unnoticed. By "bursting bubbles" that isolate information, our method and online application reveal the interconnectedness of geosciences research, supporting collaboration beyond traditional boundaries.

*Code and data availability.* The code and data for computing the embedding, comparing similarity, and creating the figures for this publica-
135 tion are documented in https://git.ufz.de/sodoge/egu-abstract-embeddings

An interactive visualization of the data is available at https://taiscarvalho.github.io/egu-umap-viz/

## Appendix A

### A1

*Author contributions.* JS: conceptualisation, methodology, investigation, data curation, formal analysis, writing (original draft preparation,
review, and editing), project administration, visualisation. TNC: conceptualisation, methodology, visualization, data curation. MMdB: conceptualisation, writing (original draft preparation, review and editing).

*Competing interests.* The contact author has declared that none of the authors has any competing interests.

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
