# Peer review of "GC Insights: Breaking the silos – leveraging NLP to encourage interdisciplinary interaction at the EGU"

_EGUsphere, 2024_

## Author Response (AR1)

**Peer review responses to comments:**

Lina Stein (RC1)

**Comment**: Your main assumption is that people will not attend presentations of research similar to their own submission due to it being in another scientific division. E.g. Line 10 "We hypothesize that this compartmentalization may inadvertently create knowledge silos, as EGU GA attendants tend to focus on their own scientific divisions, potentially missing relevant developments from other disciplines." From my own (probably biased) experience I would say that cross-division attendance does take place a lot during EGU. And the EGU website does already offer a search tool that allows people to find abstracts across scientific divisions according to keyword searches. However, text similarity offers a much better tool to identify presentations of interest. Simple keyword searches often still produce a large number of results.  While it is, of course, impossible to quantify which scientific division sessions are attended by whom during the GA, it would help your claim to offer some more numbers to demonstrate the infeasibility of manually checking all sections for relevant contributions. How many abstracts are submitted per section? How many sessions are registered per section?

**Answer**: This is a good suggestion. Indeed, as you point out our initial assumption is very stylized and might not reflect reality at EGU GA. In the updated manuscript, we added to this hypothesis that it is likely not reflecting reality yet that such simplification is necessary for our assessment. This is a limitation of our study that requires clearer acknowledgment. In fact, for the next EGU, we plan to survey individual users' attributes (e.g., which division(s) does one belong to) to assess the quality of recommendations and select the most appropriate model. Regarding the current manuscript, we agree and will implement your suggestion of adding some numbers on the volume of abstracts to emphasize the 'infeasibility of manually checking'.

**Comment**: On thing I found irritating is that the 22 scientific divisions or scientific sections are referred to as sessions. There are multiple sessions organised within each scientific division. But the authors split their data into scientific divisions and called them session. That the selection on the EGU GA website offers "Disciplinary sessions" is not a name, but means that they are the session related to one specific division.

Please clarify in the abstract and text what you mean by participant exploration or participant focus. It should be stated early on and more clearly, that you refer to choice of session/division attendance during the GA. (And not, for example, choice of session during the abstract submission process).

**Answer**: We agree with your criticism of the unclear terminology. We used the disciplinary sessions throughout the manuscript as it is the name on the GA website but understand that "scientific division" is a much clearer terminology. As such, we revised

the entire manuscript and clarified in the text that we use the term "scientific divisions" to refer to what sometimes is called disciplinary sessions.

**Comment**: Lastly, it would be nice to add the number of people who have used the webtool to find relevant sessions during EGU24.

**Answer**: Sadly, we have no clear evidence of the number of users based on the platform and subscription we used for hosting this app (shinyapps.io). The only metric we could quantify is 45 hours of usage time. However, this does not correspond to a particular number of users. For the upcoming EGU25 though, we plan to quantify such metrics more in-depth by hosting the application via our computational resources (see also the first comment on our ambitions to quantify more aspects to improve the evaluation).

**Kirsten v. Elverfeldt (RC2)**

**Comment**: Dear authors, thank you for submitting your interesting manuscript to Geoscience Communication! I enjoyed reading it and think that your use of NLP to reduce the risk to be inadvertently caught in knowledge silos when participating at the EGU GAs can be very useful. I do not want to repeat the points raised in previous comments (e.g. mixing up sessions with programme groups, the inherent inter- and transdisciplinary character of some programme groups that you might have not regarded, the existence of co-organized sessions, or that your claim that attendees focus on "their "programme group only should rather be framed as hypothesis than as a fact). Instead, I will focus on your methods and data section.

**Answer**: Thank you for your positive feedback. Regarding the issues you mention, we will address them in the updated manuscript. First, by adding the statistics for the interdisciplinary sessions to the results section (see comment by Maria-Helena Ramos). Here, we will add the number of potential presentations missed and add a note that these statistics for the interdisciplinary sessions need to be treated differently compared to the 'traditional' sessions/divisions. Second, adjusting the terminology for programme groups, sessions and divisions as described in the response to Lina Stein: we now differentiate between scientific divisions and sessions. We use the term scientific division to cover what you refer to here as program groups to provide a more intuitive wording for readers. In the introduction, we now define that scientific divisions and program groups can be seen as overlapping concepts/synonymous. Third, we will add a new paragraph in the results on the co-organized sessions and how they can impact a lower share of 'missed' presentations. Fourth, concerning the attendees' focus on "their" programme group/division, we acknowledge, in line with the other two comments, that this assumption is highly stylized (yet was required as an initial assumption for the presented analysis). As we lack the empirical evidence on participant behavior, we add to the updated manuscript that there are other potential behaviors, such as those suggested by Maria-Helena, and that our assumption here is very stylized and likely does not reflect reality. We also add that, therefore, in future research, we aim to account for and measure such conference participant behavior to provide more accurate assessments of reality.

**Comment**: I am not a NLP specialist at all, but maybe especially because of that I found your method description too intransparent. Which pre-processing steps have been applied, and how can they affect the results? Exactly which NLP-technique has been used (e.g. TF-IDF)? Do you plan to validate your results, e.g. by providing evidence in the future that users actually benefit from the NLP-produced recommendations? Furthermore, for people like me who are a layperson with respect to NLP, a flowchart of the NLP pipeline could be helpful to understand the method.

**Answer**: Thank you for bringing up this point and your perspective which helped us to think again about making the methodological procedure more explicit. To tackle this, we added an overview figure in the Appendix (adding it to the main text is not possible as only one figure is allowed). Also, we adjusted the method description in this regard. We added an initial statement on the overall procedure ("Our approach consists of 4 steps,

which we outline in detail in this section: (1) collecting abstracts from EGU, (2) computing similarities between abstracts, (3) visualizing the presentations in a 2-dimensional space, and (4) running a simulation to estimate the hypothesized filter effect…") and more detailed information on the required pre-processing steps. Concerning the validation, we added to the discussion that we plan to account for user feedback in future research which will be valuable to improve our recommendations.

[Figure]

**Maria-Helena Ramos (CC1)**

General comments:

1. **Comment**: Scientific sessions in the EGU GA are structured around disciplinary Programme Groups (PG), complemented by the EOS (Education and Outreach Sessions) programme group and the ITS (Inter- and Transdisciplinary Sessions) programme group, as they are called. Below I indicated where I think this organization should be clarified to avoid misunderstanding with the terminology. In several instances, I believe that "session" is misused in the place of "programme group". They are indicated in details below.

    **Answer**: This is correct. We have updated the terminology accordingly in the revised manuscript. In line with the comments from Reviewer 1 (Lina Stein), we now differentiate between scientific divisions and sessions. To enhance clarity for readers, we use the term scientific division instead of program groups. In the introduction, we define scientific divisions and program groups as overlapping concepts or synonymous, at least for the scope of this analysis. From the perspective of an EGU GA participant, we believe that scientific division is a more intuitive term. We only briefly responded to the follow-up, more specific comments on particular parts in the manuscript regarding this terminology issue as we believed the described changes here and in response to Lina Stein, too, are sufficient.

2. **Comment**: While I understand the choice of not considering sessions in the ITS programme group (lines 33-36), I think some clarification may be given also in relation to the fact that some "disciplinary" programme groups are, in fact, highly "inter- or transdisciplinary" too. This is, to a high degree, notably the case of NH (Natural Hazards) and Earth and Space Science Informatics (ESSI), for instance, and, maybe to a lower degree, to some other disciplinary programme groups. How could this issue affect your analysis and results?

    **Answer**:  We understand that the decision not to consider the ITS program group is debatable and we debated this as well. In the submitted version of the manuscript, we initially decided not to integrate these sessions. However, we will include them in the updated manuscript.

    Our dataset contains 3,663 presentations associated with the ITS group. For the updated results, we added ITS sessions to their associated program groups (e.g., ITS and NH). We then repeated the experiment to evaluate whether the share of missed talks had changed. Our results revealed that adding ITS sessions actually increased the number of missed relevant presentations in some divisions (AS, CR, BG, HS, SSS), but it did not reduce the number of missed presentations in any division.

    We attribute this to a potential mechanism, which we will include alongside the results in the manuscript. Because ITS sessions are interdisciplinary, their

thematic coherence is often lower than that of specific divisions. As a result, selected presentations from ITS sessions have a lower chance of matching relevant talks within their associated division. Despite this potential explanation, ITS sessions still offer valuable additional interdisciplinary perspectives which we will emphasize next to those results.

Yet, these additional findings should still be interpreted with caution, taking into account your comment 3 (see below) and our response, which will also be incorporated into the updated manuscript.

3. **Comment**: Also, I was wondering how co-organization of sessions within two or more programme groups were considered in the analysis, and how this might affect the results. Have you selected abstracts only from sessions that are not co-organized by other PGs? (session co-organizations are marked as such in the EGU GA programme: "CR3.3 - Advances in sea-ice modelling: developments and new techniques. Co-organized by NP1/OS1". In this example, the session is led by the CR programme group, but since it is co-organized it is also displayed in the NP and the OS programme groups). Programme groups have different percentage of co-organized sessions (over the total number of led sessions), which can vary significantly.

   **Answer**: This is important because we did not clarify this enough in the manuscript, as we realized some shortcomings in our data collection when you pointed them out. When collecting data for the analysis, we did not store information on the different program groups/divisions leading the session versus those organizing it. Instead, we only have information about the group leading the session (e.g., in your example, CR). Hence, in our analysis, we can only consider which program group/division leads the sessions, which is a clear limitation.

   We will add this limitation to the manuscript, highlighting that, as you mentioned, the share of such co-organized sessions varies by session and might influence the results. An important note is that we still consider the sessions with ITS (see previous comment).

4. **Comment**: When looking at Fig. 1, I was also wondering how much the very different sizes of the PGs may affect the results. The EGU programme has PGs with 2,000+ abstracts in 200+ sessions, as well as PGs with 200+ abstracts in 10+ sessions. Would this affect the analysis and results? How/why not?

   **Answer**: This is an important consideration, and we looked into it. We found a correlation of 0.38 between the number of relevant talks in the same division and the number of presentations per division. This result indicates a low correlation, where divisions with more presentations in total also have a higher share of relevant presentations among the most similar ones for each abstract.

   A possible explanation is that with more presentations, there is a higher chance of finding a similar presentation within the larger "cloud" of presentations—using the language of the cartography of presentations we created in the main figure. However, since the correlation is <0.4, this mechanism does not seem particularly

strong, meaning our results are unlikely to be heavily influenced by it.

Nonetheless, we will add this to the results section as an important consideration regarding how division size might affect the metric of missed presentations.

5. **Comment**: If I understand well, the paper suggests that using an automatic k-clustering approach could lead to more topically-similar programme groups at the EGU GA. While this might be a good idea, I think one has to consider also the "errors" of the text similarity approach. Maybe adding some comments on that in the Discussion section would be important here. Also, one has to consider that, currently, mostly it is up to the authors to make the decision of where they want to submit their abstract to. From my experience, I believe this choice depends on the disciplines, but also on the community around a given session (conveners team, authors of the same session in the past year(s), etc.). Under a clustering approach as proposed, this might be lost, if I understand well, and the system will make this choice for the author. In your opinion, what would be the obstacles/limitations here?

   **Answer**: When discussing the use of k-means for organizing sessions at EGU, we were careful not to propose a purely text-based approach. We fully acknowledge the limitations of text similarity methods, including potential classification errors and the challenge of capturing disciplinary nuances. Your point about the role of authors in selecting the most appropriate session—often influenced by community ties, and past participation—is particularly important.

   We do not yet have a clear roadmap for how k-means clustering could be implemented. Most likely, it would need to be integrated through a co-design process to ensure it complements, rather than replaces, the existing system. Our initial framing was intentionally somewhat provocative to stimulate discussion on these complexities. In the revised manuscript, we will clarify this perspective and explicitly address its limitations, as well as the need to preserve author agency in session selection.

6. **Comment**: In the Discussion section, it could be interesting to add that the authors are currently working with the EGU Programme Committee and Copernicus Conference Manager to experiment with additional tools that could be integrated to the conference system and improve the experience attendees and authors may have in future EGU General Assemblies. These experiments can be useful to validate the NLP algorithms in real case applications.

   **Answer**: Yes, this is an important aspect to highlight concerning the actual implementation of such more vague and theoretical thought experiments. We will highlight both the example of the tool used for recommending participants sessions to submit their abstracts to and the tool for recommending relevant presentations for participants during the conference. For the latter, as mentioned in the previous peer review comments, we will highlight how this can help for systematically incorporating user feedback.

**Specific comments:**

**General suggestion**: replace "attendants" to "attendees"

**Answer**: Accepted

**Line 7**: EGU General Assembly gathers over 20,000 participants worldwide. I suggest to update the numbers here.

**Answer**: Accepted

**Lines 8-9**: change to: "…the EGU GA is currently structured into 22 disciplinary programme groups (PGs) and further sessions included in the ITS (Inter- and Transdisciplinary Sessions) programme group and the EOS (Education and Outreach Sessions) programme group, among other Union-wide sessions. We hypothesize…"

**Answer**: Accepted

**Line 10**: change to: "…as EGU GA attendees might tend to focus on the disciplinary PG related to their own scientific discipline, potentially…" => I believe that "might" should be used here as, to my knowledge, so far there is no study that has put this into evidence.

**Answer**: Accepted

**Line 22**: change to: "… the compartmentalization at the EGU GA programme presentation may…" => since there are instruments for co-organizations as well as other Union-wide sessions in the EGU GA programme, I believe that a precision must be added here to say that it refers to the EGU GA programme presentation (not to the EGU GA as a whole).

**Answer**: Accepted

**Line 30**: do you mean "corresponding session identification (session ID)"? I think "session data" is a bit unclear.

**Answer**: Accepted

**Line 31**: change "presented in 22 disciplinary sessions" to "presented in the 22 disciplinary programme groups", and Line 31-32: change to "disciplinary programme groups" (or "disciplinary PGs") => I think you are referring to PGs, rather than sessions. "Sessions" are the individual sessions inside the PGs, and there are more than 22 of those at EGU GA (usually, there are 1,000+ sessions in an EGU GA).

**Answer**: This has been adapted in light of the comment from Lina Stein regarding the new terminology for scientific divisions and sessions.

**Line 32**: change to: "…may vary significantly…"

**Answer**: Accepted

**Line 33**: suggestion to change to: "… identify their research as "interdisciplinary",

**Answer**: This part has been deleted in light of the major comments made on the exclusion of the ITS sessions.

**Line 34**: I do not fully understand this sentence: could you clarify? I am not sure that "rarely align with ITS sessions" may be fully correct. The ITS sessions are very popular among authors, and the number of abstract submissions to the ITS PG has well increased over the years. Also, it is not clear to me what you mean by "tend to be… narrowly focused", since the nature of the ITS PG sessions is broad, involving one or more disciplines.

**Answer:** This was indeed phrased as not ideal by us here as it causes some misunderstandings. The 'narrowly focussed' we intended to say that the contributions in these sessions often do not overlap with other divisions also considering how they change between the years. We acknowledge that 'narrowly focussed' is the wrong description here. We will adjust this sentence where we discuss the ITS based on your previous comment above.

**Line 35**: change to: "… concentrating only on abstracts submitted to sessions within disciplinary programme groups provides…"

**Answer**: This part has been deleted in light of the major comments made on the exclusion of the ITS sessions.

**Lines 50-53:** I got a bit confused here: do you mean "sessions" or "programme groups"? I have the impression that you mean "programme groups" (at list on lines 52 and 53); please check (see also my general comment above)

**Answer**: As per our comments earlier, this has been adjusted to sessions and scientific divisions.

**Lines 54-55**: change to: "…to evaluate how well the grouping of the EGU GA abstracts in the 22 PGs compares with the clustering using the k-means clustering algorithm,

considering the same number of clusters as the number of disciplinary PGs (n=22)."
Please, pay attention to the fact that there may be a terminology-based confusion here also between "sessions" and "programme groups".

**Answer**: As per our comments earlier, this has been adjusted to sessions and scientific divisions.

**Line 60**: suggestion to change to "…landscape of the EGU GA abstract sample analysed here, …"

**Answer**: Accepted

**Line 61**: change "disciplinary session" to "disciplinary PG"

**Answer**: As per our comments earlier, this has been adjusted to sessions and scientific divisions.

**Line 62**: change "… and climate (CL) session…" to "… and climate (CL) programme groups..."

**Answer**: As per our comments earlier, this has been adjusted to sessions and scientific divisions.

**Line 64**: change "disciplinary sessions" to "disciplinary PGs"

**Answer**: As per our comments earlier, this has been adjusted to sessions and scientific divisions.

**Lines 64-65**: suggestion to change to "… to a research abstract on modelling… include an abstract on evaluating…"

**Answer**: Accepted

**Lines 67**: I think here also you mean "presented in different programme groups", not different "sessions"; is that so? Or maybe "presented in different sessions, belonging to different programme groups" (?)

**Answer**: Your are right, has been changed to scientific division.

**Line 67**: Maybe complete with the following: "… would potentially be missed if an attendee only consults one programme group and the sessions are not co-organized". => I think this is important to mention since consulting other PGs when preparing a "personal programme" and displaying co-organized sessions in all the programmes of

the PGs involved in the co-organization (as done in the EGU GA programme) would minimize this effect.

**Answer**: We agree that this is a good opportunity to remind the reader of our initial hypothesis so we added this to the manuscript here.

**Line 68**: change to "disciplinary programme groups"

**Answer**: As per our comments earlier, this has been adjusted to sessions and divisions.

**Line 69**: also here, do you mean "within the programme group where their abstract was submitted to potentially…"? Instead of "session"?

**Answer**: As per our comments earlier, this has been adjusted to sessions and divisions.

**Line 70**: Suggestion: I would rather say "…. of the 10 contributions most similar to their abstract".

**Answer**: accepted

**Line 70 and Line 72**: change "session" to "programme group" in the three occurrences in these lines. AS and ST, as well as GI and NP, are not sessions, but "programme groups".

**Answer**: As per our comments earlier, this has been adjusted to sessions and divisions.

**Lines 73-74**: I think the sentence is a bit confusing (I could not grasp the message/ meaning). Maybe rephrase it? Sentence: "…reduce the share of relevant contributions covered by the own "session=> programme group"

**Answer**: Changed to "have a lower likelihood for attendees to miss relevant presentations"

**Line 75:** change to "using disciplinary programme groups"

**Answer**: As per our comments earlier, this has been adjusted to sessions and divisions.

**Line 77-78**: I am bit confused here: do you mean to group "similar abstracts"? I do not understand the term "group sessions". Sessions are grouped in programme groups. Maybe you mean: "to build programme groups that instead of focusing on a discipline would bring together statistically similar abstracts".

**Answer**: changed to "This suggests that using a statistical clustering approach to group abstracts produces significantly more coherent clusters compared to the divisions (and programe groups)."

**Line 81**: the sentence may be missing something: to keep track of what exactly? Have you consulted participants on that? And organizers (what type of organizers?

Conveners? Programme Group chairs?). I think it might be useful to provide more details here.

**Answer**: keeping track "...of relevant presentations" has been added. As we added to the discussion, so far these have been based on assumptions and anecdotal evidence. This is why we added for a perspective on future research that we aim to track participant behavior in more detail. See also your comment and our answer line 84.

**Line 82**: replace by "last five General Assemblies"

**Answer**: accepted

**Line 84:** suggestion: "the participant identifies most with". Having said that, I believe that the issue is not only "which discipline a participant identifies with", but how participants search the programme of the GA to prepare their own personal programme: do they stay in the same programme group or do they navigate through the other programme groups as well? As the number of abstracts increase, I agree that searching through the programme, inside the programme groups and their respective sessions, might become a laborious task tough!

**Answer**: The laborious task of searching for presentations is indeed a competing or complementary explanation. We add that both are valid reasons in the updated manuscript. As we do not have empirical evidence on the strength of each effect, it is important to highlight all potential mechanisms.

**Line 85**: suggestion: "the possible presence of a bubble effect" => to me, a bubble effect in itself needs more than a disciplinary-based programme display organization; it requires also that participants do not look outside their disciplinary programme group, for instance.

**Answer**: This is a comment we wanted to pick up on in some more detail because it makes a valid point that we will underline in the updated manuscript. Concerning the bubble effect, we do not posit that such an effect exists as it is difficult to quantify without empirical data on conference participant behavior anyway. We stressed this assumption again also following the first peer review comment by Lina Stein on this issue.

**Line 86**: replace "sessions" to "programme groups" or "disciplines"

**Answer**: As per our comments earlier, this has been adjusted to sessions and divisions.

**Line 87**: could it be related to the fact that in EGU GA NP and GI are also smaller programme groups, comparatively to PGs such as AS and HS, for instance?

**Answer**: see our previous comment on the correlation which we added to the manuscript.

**Line 88**: replace "sessions" to "programme groups"

**Answer**: As per our comments earlier, this has been adjusted to sessions and divisions.

**Line 94**: replace "session" to "programme group"

**Answer**: As per our comments earlier, this has been adjusted to sessions and divisions.

**Line 95**: change to :.. interdisciplinary connections, which in turn may lead to innovative…". Also, I don't understand the "higher productivity" here? Do you mean (even) more research papers being published? Projects funded? Please, explain.

**Answer**: Thank you for pointing this vague phrasing out. We decided to remove this part from the sentence as it is very vague and more specific outcomes such as funding are not specified in the literature either.

**Lines 96-97**: change "EGU GA 2024" to "EGU24 General Assembly"

**Answer**: accepted

.

**Line 98**: replace "sessions" to "programme groups"

**Answer**: As per our comments earlier, this has been adjusted to sessions and divisions.

**Line 100**: replace "sessions" to "programme groups"

**Answer**: As per our comments earlier, this has been adjusted to sessions and divisions.

**Figure 1 caption**: change "session" to "programme group" in the four occurrences.

**Answer**: As per our comments earlier, this has been adjusted to sessions and divisions.